# Decision-based Black-box Attack Against Vision Transformers via Patch-wise Adversarial Removal

**Yucheng Shi, Yahong Han**[*]
College of Intelligence and Computing, and Tianjin Key Lab of Machine Learning
Tianjin University, Tianjin, China
Engineering Research Center of City Intelligence and Digital Governance
Ministry of Education of the People's Republic of China
{yucheng, yahong}@tju.edu.cn

**Yu-an Tan**
School of Cyberspace Science and Technology
Beijing Institute of Technology, Beijing, China
tan2008@bit.edu.cn

**Xiaohui Kuang**
National Key Laboratory of Science and Technology on Information System Security, Beijing, China
xiaohui_kuang@163.com

## Abstract

Vision transformers (ViTs) have demonstrated impressive performance and stronger adversarial robustness compared to Convolutional Neural Networks (CNNs). On the one hand, ViTs' focus on global interaction between individual patches reduces the local noise sensitivity of images. On the other hand, the neglect of noise sensitivity differences between image regions by existing decision-based attacks further compromises the efficiency of noise compression, especially for ViTs. Therefore, validating the black-box adversarial robustness of ViTs when the target model can only be queried still remains a challenging problem. In this paper, we theoretically analyze the limitations of existing decision-based attacks from the perspective of noise sensitivity difference between regions of the image, and propose a new decision-based black-box attack against ViTs, termed Patch-wise Adversarial Removal (PAR). PAR divides images into patches through a coarse-to-fine search process and compresses the noise on each patch separately. PAR records the noise magnitude and noise sensitivity of each patch and selects the patch with the highest query value for noise compression. In addition, PAR can be used as a noise initialization method for other decision-based attacks to improve the noise compression efficiency on both ViTs and CNNs without introducing additional calculations. Extensive experiments on three datasets demonstrate that PAR achieves a much lower noise magnitude with the same number of queries.

## 1 Introduction

Vision transformers (ViTs)[1] not only achieve significant performance improvement in a wide range of computer vision tasks [2, 3, 4], but also show stronger robustness against adversarial examples generated by different attack methods [5, 6, 7, 8]. The adversarial examples are generated by attackers

---

[*]Corresponding author.

36th Conference on Neural Information Processing Systems (NeurIPS 2022).

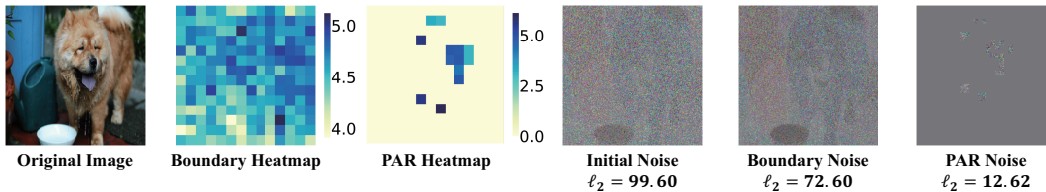

| Original Image | Boundary Heatmap | PAR Heatmap | Initial Noise $\ell_2 = 99.60$ | Boundary Noise $\ell_2 = 72.60$ | PAR Noise $\ell_2 = 12.62$ |

Figure 1: Noises of Boundary Attack and our method PAR after 100 queries from the same initial noise. Heat maps visualize the noise magnitude of each patch. PAR removes noise in patches with low noise sensitivity, achieving significantly smaller noises than Boundary Attack.

to fool the target model by adding imperceptible noises to original data [9, 10, 11]. The characteristic of using non-overlapping patches in ViTs reduces the influence of adversarial examples with the same noise magnitude on the overall results [5].

According to the amount of information the attacker can obtain, adversarial attacks can be divided into white-box attacks and black-box attacks [12]. In the black-box scenario where the attacker can only obtain the hard labels output by the target model, adversarial attacks can be further divided into transfer-based methods [13, 14] and decision-based methods [15]. Transfer-based methods use a substitute model to generate adversarial examples and transfer them to the target model taking advantage of the transferability [16, 17, 18]. Compared with transfer-based attacks, decision-based attacks face a more practical setting where a substitute model is not available. The only source of information for decision-based attacks is hard label obtained by querying the target model. In the image classification task, the decision-based attacks [15, 19, 20] start from a random noise with a large noise magnitude, randomly sample in the image input space, and gradually compress the adversarial noise under the premise of ensuring misclassification. The existing adversarial attacks against transformers are only white-box attacks [21, 5, 22, 23] and transfer-based black-box attacks [24, 25]. The characteristics of ViTs' patch-wise splitting of images reduce the impact of adversarial noise, leaving decision-based black-box attacks against ViTs an open problem [5].

The challenge of attacking ViTs with decision-based methods comes from their properties in noise sensitivity, which derive from the structural characteristics of ViTs. On the one hand, ViTs learn fewer low-level features and more transferable features than CNNs, resulting a much more noise needed to attack ViTs [21]. In other words, the overall noise sensitivity of ViTs is low. Decision-based attacks against ViTs need to add random noise with much larger noise magnitude to find the initial adversarial examples. Larger initial noise makes it more difficult for the decision-based attacks to compress, i.e., to find the smallest adversarial noise under the same number of queries. On the other hand, ViTs split the image into multiple non-overlapping patches, which reduces the impact of noises on one single patch to the final classification results [5]. This leads to a notable difference in ViTs' noise sensitivity between different regions of an image, which is rarely considered by existing decision-based methods. For example, the noise compression process of Boundary Attack [15] treats all pixels equally regardless of their noise sensitivity, as demonstrated in Fig. 1, which severely hinders the efficiency of noise compression. These two properties in noise sensitivity make it extremely difficult for existing decision-based attacks to find adversarial examples with small noise magnitude against ViTs. Noise sensitivity directly reflects the black-box adversarial robustness of ViTs, which has not been well studied and unable to shed light on the mechanism of improving noise compression efficiency. Therefore, designing decision-based attacks against ViTs according to their noise sensitivity properties is an essential problem.

In this paper, we verify that noise sensitivity of ViTs varies significantly between different image regions. The limitation about the compression process of decision-based attacks represented by Boundary Attack is theoretically analyzed. Based on the relationship between ViTs' patch-wise sensitivity and noise compression success rate, we propose a new decision-based attack method Patch-wise Adversarial Removal (PAR). PAR splits the adversarial example into multiple patches and perform coarse-to-fine noise removal. Specifically, PAR maintains two masks to record the noise sensitivity and noise magnitude of each patch, respectively. Before querying the target ViTs, PAR locates the patch with the highest query value based on these two masks. As the search progresses, the size of each patch gets smaller while the measure of the noise sensitivity of the ViTs becomes

more accurate. PAR achieves a significant noise compression under a small number of queries, which can be used as an initialization for other decision-based attacks without additional computation.

We validate the effectiveness of PAR on three datasets: ImageNet-21k [26], ILSVRC-2012 [27], and Tiny-Imagenet [28]. We compare PAR with 7 state-of-the-art decision-based attacks against 18 different target models, including 8 CNNs and 10 ViTs or hybrid models. Benefiting from the powerful redundant noise compression capability, the noise magnitude of all decision-based attacks has been notably reduced after using PAR for noise initialization without increasing the query number.

## 2 Related Work

### 2.1 Robustness of Vision Transformer

The ViTs show stronger adversarial robustnes[29]. Not only does fooling ViTs in the white-box scenario require larger noise magnitude [21], but it is also difficult for existing transfer-based black-box attacks to transfer adversarial examples from CNNs to ViTs [24]. Existing research mainly focuses on the white-box attacks and transfer-based black-box attacks against ViTs. However, this paper explores the decision-based attack against black-box ViTs without substitute model.

### 2.2 Decision-based Attack

Decision-based attacks do not rely on substitute models, but require an initial adversarial example that has already been misclassified as starting point. Boundary Attack [15] starts from an gaussian noise and searches along two directions simultaneously, namely source direction and spherical direction:

$$x^*_{new} = x^* + \delta \cdot \frac{\eta}{\|\eta\|_2} + \varepsilon \cdot \frac{x - x^*}{\|x - x^*\|_2}, \quad \eta \sim \mathcal{N}(0, I) \tag{1}$$

where $x^*$ is the adversarial example with smallest noise that already been found. $\eta$ and $(x - x^*)$ refer to the directions of spherical and source direction, respectively. $\delta$ and $\varepsilon$ are stepsizes of spherical and source direction. The Biased Boundary Attack [20] concentrates on low-frequency domain of input space to make the adversarial example more 'natural'. The Evolutionary Attack [19] reduces the dimension of sampling space by bilinear interpolation. Evolutionary attack performs better in tasks involving strong prior knowledge such as face recognition. HopSkipJumpAttack [30] estimates the gradient direction using binary information at the decision boundary. Customized Adversarial Boundary (CAB) [31] uses current noise to select the sensitive regions of images and customizes sampling distribution. SurFree Attack [32] is based on the geometrical mechanism to get the biggest distortion decrease for a given direction to be explored. Sign-OPT attack [33] uses a zeroth order oracle to compute the sign of directional derivative of the attack objective.

Boundary Attack maintains two search directions according to Eqn. (1). The source direction is pointed to the original image $x$ with a small stepsize, which is responsible for noise compression. The spherical direction is a random direction with a large stepsize, which is used to perturb in the neighborhood of current adversarial example $x^*$ and ensure that the updated adversarial example is still misclassified by the target model. It can be seen from Eqn. (1) that in the one-step update of the current adversarial example, the spherical direction $\eta$ follows the standard Gaussian distribution with the same dimensions as the original image. The source direction points to the opposite direction of the current adversarial noise. Both search directions show no preference for any region or pixel in the image. In other words, under a uniform initial random adversarial noise, the Boundary Attack performs noise compression with basically the same magnitude for all pixels. In fact, the Boundary Attack does not make any distinction between different pixels in the image, and the noise compression of each pixel is proportional to its initial noise magnitude [31].

## 3 Proposed Method

### 3.1 Notation

Suppose $F$ is the target model to be attacked: $F : X^N \rightarrow Y^C$, where $X$ represents the input space, $N$ is the dimension ($N = Width \times Height \times Channel$ for image data) and $Y$ represents the

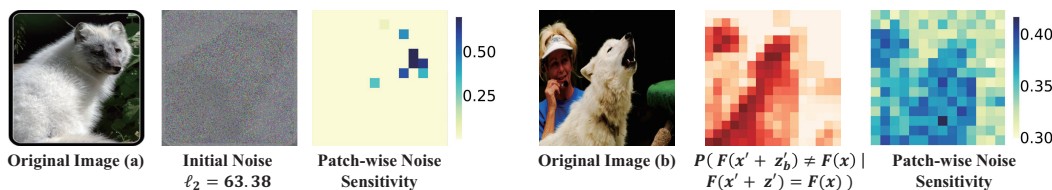

Figure 2: Illustrations of original images, initial random noises, and corresponding visualizations of patch-wise noise sensitivity.

classification space with $C$ categories. The goal of decision-based attack can be expressed as:

$$\min_{x' \in S_Q} \|x' - x\|_v, \ \ s.t. \ F(x') \neq y \ and \ |S_Q| \leq T, \tag{2}$$

where $x$ represents the original image, $x'$ refers to adversarial example, $y$ is the label of $x$, $S_Q$ is the set of all adversarial examples generated to query the target model, $T$ is the limit of query number. $v$ refers to the norm used to measure the noise magnitude including $\ell_1$, $\ell_2$ and $\ell_\infty$ norm. In this paper the $\ell_2$ distance is calculated. In the process of decision-based attack, the attacker can only obtain the hard label $F(x')$ output by the target model.

### 3.2 Noise Sensitivity of ViTs

Here we measure the noise sensitivity $Sens$ of one patch in the image according to the maximum ratio the noise can be compressed under the current adversarial example on ViT:

**Definition 1.** *Let $x'$ be an adversarial example of ViT model $F$ on the original image $x$, i.e., $F(x') \neq F(x)$, and $z$ be the current adversarial noise $z = x' - x$. Let $\tilde{z}$ be a new adversarial noise compressed from $z$ in a rectangle patch with width of $w$, height of $h$, top left corner of $sr, sc$:*

$$\tilde{z}(sr, sc, h, w, \kappa)_{r,c} = \begin{cases} z_{r,c} \cdot \kappa, & if \ sr \leq r < sr + h \ and \ sc \leq c < sc + w, \\ z_{r,c}, & else, \end{cases} \tag{3}$$

*where $r$ and $c$ refer to the row and column index of one pixel in noise $z$, respectively. $\kappa \in [0, 1]$ denotes the noise compression ratio. Define the noise sensitivity of a rectangle patch as the minimum noise compression ratio $\kappa_{min}$ when $F$ misclassifies $x + \tilde{z}$:*

$$\begin{aligned} Sens(F, x, x', sr, sc, h, w) = \kappa_{min}, \ \ s.t. \ \ & F(x + \tilde{z}(sr, sc, h, w, \kappa_{min})) \neq F(x) \\ and \ \forall \ \kappa' < \kappa_{min}, \ \ & F(x + \tilde{z}(sr, sc, h, w, \kappa')) = F(x). \end{aligned} \tag{4}$$

$Sens$ measures the minimum amount of noise that is required for an adversarial example. A smaller $Sens$ means that more noise can be removed without changing the misclassification results, i.e., adding noise in this patch has less impact on classification results. When $h = w = 1$, $Sens$ measures the pixel-level noise sensitivity.

We use a vision transformer vit-tiny-patch16 [34] with patch size of 16 trained on ILSVRC-2012 [27] as the target model. To demonstrate the difference of patch-wise noise sensitivity in Fig. 2, we add initial Gaussian noise to the original images until they are misclassified. After getting the initial noise, we try to reduce the noise on each patch to evaluate the patch-wise noise sensitivity of the images on the target model. Since the size of the original image is $224 \times 224 \times 3$, there are $14 \times 14$ patches. We use binary search to evaluate $Sens$ on each patch:

$$L = x^{init}, R = x^{init}, L_{row*16+1:(row+1)*16, col*16+1:(col+1)*16} = 0, \tag{5}$$

$$BS(L, R) = \begin{cases} BS(L, (L+R)/2), if \ F((L+R)/2) \neq y, \\ BS((L+R)/2, R), if \ F((L+R)/2) = y, \end{cases} \tag{6}$$

where $row, col \in [1, 14]$ refer to the row index and column index of one patch in the image, respectively. The heat map in Fig. 2 shows the $Sens$ on all $14 \times 14$ patches. In the heat map, a lighter color means lower $Sens$ on one patch. 1 indicates that any small noise compression on this

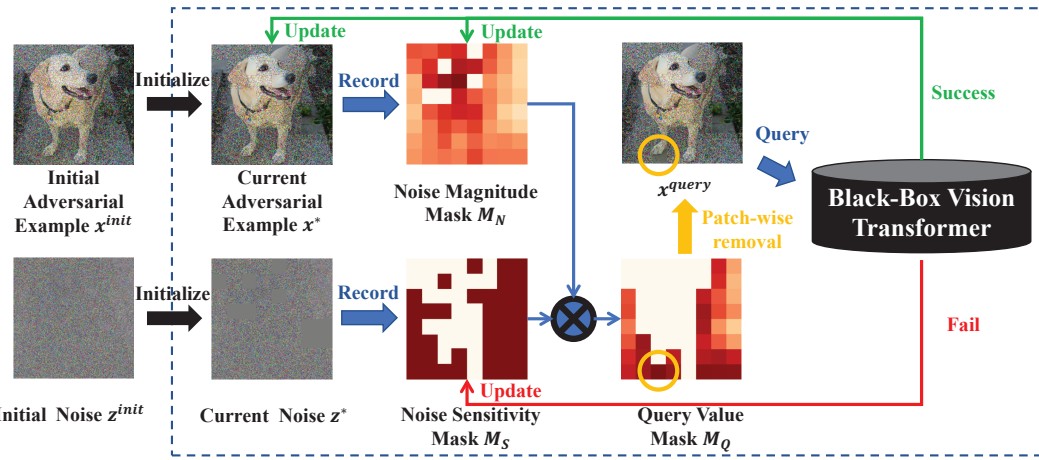

Figure 3: The noise compression process of PAR. Based on the initial adversarial example, PAR records the noise magnitude of current adversarial example and the noise sensitivity from historical queries, respectively. After locating the patch with the highest query value (yellow circle) using the product of two masks, the noise on the corresponding patch of current adversarial example is removed and query the target ViT. If misclassified, PAR updates the noise magnitude mask. Otherwise the corresponding patch on the noise sensitivity mask is set to zero.

patch makes the target model outputs correct label. 0 indicates that the adversarial example remains misclassified even all noises on this patch are removed. Considering the attack process of Boundary Attack, the patch-wise sensitivity of adversarial examples and the compression success probability within a patch of one single query have the following relationship:

**Proposition 1.** *Assume $x'$ is an initial adversarial example generated by Boundary Attack against ViT $F$ starting from original image $x$, $F(x) \neq F(x')$. For any $0 < r_1, r_2, h \leq Height$, $0 < c_1, c_2, w \leq Width$, if $Sens(F, x, x', r_1, c_1, h, w) < Sens(F, x, x', r_2, c_2, h, w)$, and the new noise added by one step by Boundary Attack is $z'$, then $P(F(x' + z'_1) \neq F(x)|F(x' + z') = F(x)) < P(F(x' + z'_2) \neq F(x)|F(x' + z') = F(x))$, where for $\iota = 1, 2$*

$$z'_{\iota,r,c} = \begin{cases} 0, & if \quad r_\iota \leq r < r_\iota + h \quad and \quad c_\iota \leq c < c_\iota + w, \\ z'_{r,c}, & else, \end{cases} \tag{7}$$

*Proof Idea.* The expectation of each pixel on the initial noise $x'$ generated by Boundary Attack is equal, and the noise compression ratio after one-step Boundary Attack for each pixel is i.i.d. The possibility that the noise compression ratio on at least one pixel exceeds $\kappa$ is also the same for any patch. Since the probability that noise compression ratio on at least one pixel exceeds $Sens$ increases monotonically w.r.t. the noise sensitivity on the whole patch, the noise removal of high $Sens$ patch is more likely to be the cause of misclassification failures. □

The detailed proof of Proposition 1 is provided in the Supplementary Material A.2. Proposition 1 indicates that during the noise compression process of Boundary Attack, patches with higher $Sens$ are more likely to be the cause of query failure than regions with lower $Sens$. Obviously, the noise sensitivity of ViTs varies greatly in different regions of the image. In the right part of Fig. 2, we compare the patch-wise sensitivity of original image (b) and the probability that noise compression ratio exceeding $Sens$ on each patch $P(F(x' + z'_b) \neq F(x)|F(x' + z') = F(x))$ caused by one step of Boundary Attack. As can be seen that these two heat maps are basically consistent on patches, which verifies Proposition 1. It can be seen from Fig. 2 that completely removing noises of many patches from $x^{init}$ does not affect the misclassification. However, the uniform compression of Boundary Attack usually keeps these redundant noises to the end. The magnitude of the whole block of redundant noise is considerable, especially for ViTs which require a larger initial noise.

---

**Algorithm 1** Patch-wise Adversarial Removal

---

**Input:** Target model $F(x)$, noise magnitude limit $\tau$, original image $x$ and label $y$
 Max querying number $T$, initial variance of gaussian distribution $var$
 Identity matrix $I$ of the same dimension as $x$, initial and minimum patch size $PS_0$ and $PS_{min}$

**Output:** Adversarial example $x^*$ with compressed noise

**while** $F(x^{init}) = y$ **do**
 $\quad \xi^{init} \sim \mathcal{N}(0, var^2 I), \quad x^{init} \leftarrow Clip_{x,\tau}\{x + \xi_0^{Gau}\}, \quad var \leftarrow var * 2, \quad T \leftarrow T - 1$

Initialize $M_N$ and $M_S$ according to Eqn. (8) and Eqn. (9), $\quad PS \leftarrow PS_0, x^* \leftarrow x^{init}$

**while** $T > 0$ **do**
 $\quad M_Q \leftarrow M_N \odot M_S$
 $\quad$ **if** $\sum M_Q = 0$ **then**
 $\quad\quad PS \leftarrow PS/2$, initialize $M_N$ and $M_S$ according to Eqn. (8) and Eqn. (9)

 $\quad$ **if** $PS \leq PS_{min}$ **then**
 $\quad\quad$ break

 $\quad$ // *Locate the highest value in query value mask*
 $\quad row^*, col^* \leftarrow argmax(M_Q)$
 $\quad z^{query} \leftarrow x^* - x, \quad z_{row^**PS+1:(row^*+1)*PS, col^**PS+1:(col^*+1)*PS}^{query} \leftarrow 0$
 $\quad x^{query} \leftarrow Clip_{x,\tau}\{z^{query} + x\}$
 $\quad$ **if** $F(x^{query}) \neq y$ **then**
 $\quad\quad x^* \leftarrow x^{query}$, update $M_N$
 $\quad$ **else**
 $\quad\quad M_{S_{row^*, col^*}} \leftarrow 0$
 $\quad T \leftarrow T - 1$

---

### 3.3 Patch-wise Adversarial Removal

According to Proposition 1, the Boundary Attack compresses the overall noise of $x'$ together, whose noise compression rate depends on those patches with the highest $Sens$. Ideally, a decision-based attack should firstly compresses regions with low $Sens$ and high noise magnitude. In this way, both the success rate of query and the magnitude of one-step noise compression can be guaranteed, and the efficiency of noise compression can be maximized with limited query numbers. On the one hand, the noise sensitivity of the target model to different regions of the initial noise cannot be directly obtained. On the other hand, using binary search similar in Fig. 2 for one patch and perform grid search for all the patches consumes large quantity of queries. Therefore, we propose a new decision-based attack Patch-wise Adversarial Removal (PAR). PAR divides the initial noise into patches, probes their noise sensitivity and compresses the noise in a patch-wise manner.

As illustrated in Fig. 3, PAR guides the probe process by maintaining two masks that record the noise sensitivity of the target model and the noise magnitude of each mask, respectively. Since the detail of the ViTs is not available in the black-box attack setting, PAR does not assume the patch size of ViTs, but starts from a large patch size to conduct multiple rounds of coarse-to-fine search.

Firstly, PAR initializes the noise sensitivity mask $M_S$ and noise magnitude mask $M_N$. The shape of the two masks is $PS_0 \times PS_0$, where $PS_0$ is a hyper parameter denotes the initial patch size of PAR. We use the initial noise magnitude in $\ell_2$ of each patch in $x^{init}$ to initialize $M_N$:

$$M_N(row, col) = \sqrt{\sum_{i=row*PS_0+1}^{(row+1)*PS_0} \sum_{j=col*PS_0+1}^{(row+1)*PS_0} (x_{i,j}^{init} - x_{i,j})^2}, \tag{8}$$

where $row$ and $col$ indicate the row index and column index of $M_N$, $row, col \in [1, PS_0]$. The noise sensitivity mask is binary. 1 in $M_S$ indicates that the corresponding patch in the adversarial noise remains a low noise sensitivity state, which may not have been queried or noise removal has been successfully carried out. 0 in $M_S$ indicates that the previous noise compression process is failed. The initial value for each element in $M_S$ is 1. Before each query to the target model, we use element-wise product to obtain the query value mask $M_Q$:

$$M_S = J_{row,col}, \quad M_Q = M_N \odot M_S, \tag{9}$$

where $J$ is a unit matrix of all-ones.

Table 1: Median and average $\ell_2$ distance of adversarial perturbations on Tiny-Imagenet.

| Target | res-18 | | inc-v3 | | inc-res | | nasnet | |
|---|---|---|---|---|---|---|---|---|
| Methods | median | average | median | average | median | average | median | average |
| Initial | 2.542 | 5.024 | 8.238 | 8.402 | 10.255 | 9.933 | 8.853 | 8.428 |
| PAR | 0.45 | 1.104 | 1.457 | 1.961 | 1.805 | 2.279 | 1.723 | 2.022 |
| HSJA | 0.959 | 2.762 | 3.479 | 4.576 | 5.053 | 5.603 | 4.226 | 5.237 |
| PAR+HSJA | 0.396 | 1.067 | 1.392 | 1.899 | 1.793 | 2.236 | 1.668 | 1.992 |
| BBA | 0.23 | 0.787 | 1.091 | 1.669 | 1.565 | 2.041 | 1.361 | 1.815 |
| PAR+BBA | 0.142 | 0.605 | 0.723 | 1.25 | 1.126 | 1.59 | **0.948** | 1.463 |
| Evo | 0.522 | 1.518 | 2.043 | 2.971 | 2.892 | 3.516 | 2.411 | 3.448 |
| PAR+Evo | 0.294 | 0.882 | 1.183 | 1.701 | 1.662 | 2.01 | 1.532 | 1.835 |
| Boundary | 0.577 | 1.194 | 1.552 | 2.091 | 2.38 | 2.807 | 1.967 | 2.388 |
| PAR+Boundary | 0.296 | 0.813 | 1.034 | 1.457 | 1.478 | 1.852 | 1.425 | 1.773 |
| SurFree | 0.143 | 0.653 | **0.627** | 1.233 | 1.126 | 1.772 | 0.963 | 1.639 |
| PAR+SurFree | **0.14** | **0.599** | 0.629 | **1.171** | **1.087** | **1.479** | 0.952 | **1.453** |
| CAB | 0.397 | 0.977 | 1.103 | 1.819 | 1.372 | 2.245 | 1.23 | 2.301 |
| PAR+CAB | 0.248 | 0.728 | 0.803 | 1.326 | 1.11 | 1.604 | 0.968 | 1.474 |
| Sign-OPT | 2.134 | 4.293 | 6.669 | 7.268 | 7.037 | 8.274 | 7.332 | 7.394 |
| PAR+Sign-OPT | 0.433 | 0.957 | 1.426 | 1.926 | 1.712 | 2.012 | 1.573 | 2.008 |

If a patch does not contain noise or previous query is failed, there is no query value. We sort the values in $M_Q$ in descending order, and remove the noise in the patch corresponding to the highest value in $M_Q$. We input the updated adversarial examples $x^{query}$ into the target model to obtain the query result. If $x^{query}$ still misclassifies the target model, it indicates that the noise sensitivity of this patch is low. In this case, we set $x^*$ as $x^{query}$ and update $M_N$. Otherwise, the noise sensitivity of the patch is high, and the corresponding element in noise sensitivity mask $M_S$ is set to 0.

If the sum of $M_Q$ is 0, all patches under current patch size $PS$ either have no noise, or the query has already been made. In this case, we halve the patch size and reinitialize $M_N$ and $M_S$ according to Eqn. (8) and Eqn. (9). The next round will conduct more fine-grained query on the patches where there are still noises. Since the noise on some patches have been removed in the previous rounds, the search process of PAR with gradually reduced patch size is much more efficient in terms of queries than using very small patch size at the beginning.

There are two exit conditions for the search process of PAR, either the max query number $T$ is reached or the minimum patch size $PS_{min}$ has been reached. The $PS_{min}$ is set for the efficiency of one-step noise compression. When the patch size is too small, the compressed noise magnitude after one step is not worthwhile even if the subsequent query succeeds. Algorithm 1 details PAR.

### 3.4 PAR as Noise Initialization Method

As an query-efficient decision-based attack, PAR can also be used as a noise initialization method for other decision-based methods. PAR removes all possible blocks of noise larger than the minimum patch size $PS_{min}$, leaving the remaining regions to be noise-sensitive for ViTs. In this way, PAR greatly reduces the search space for subsequent noise compression. After initializing noises with PAR, decision-based attacks may concentrate each sampling in the regions with higher noise sensitivity.

## 4 Experiments

### 4.1 Experiment Settings

We conduct experiments on three image classification datasets: ImageNet-21k [26], ILSVRC-2012 [27], and Tiny-Imagenet [28]. We pick 10000 images from the validation sets of ImageNet-21k and ILSVRC-2012 that can be correctly classified by all target models for test. As for Tiny-Imagenet with 200 image categories, we choose 2000 images, 10 images for each category. 10 vision transformer models with different structures [34] are compared: vit-s32, vit-b16, vit-b32, r50-l32, r50-s32, vit-large-patch16-224, vit-tiny-patch16-224, vit-small-r26-s32-224, vit-tiny-patch16-224, vit-small-patch16-224. We also include 8 CNNs as target models: resnet-18 [35], resnet-101, inception v3 [36],

Table 2: Median and average $\ell_2$ distance of adversarial perturbations on ImageNet-21k.

| Target | r26_s32 | | ti_s16 | | vit_s16 | | ti_l16 | | r_ti_16 | |
|---|---|---|---|---|---|---|---|---|---|---|
| Methods | median | average | median | average | median | average | median | average | median | average |
| Initial | 41.161 | 43.24 | 21.376 | 26.847 | 40.52 | 45.828 | 23.591 | 43.866 | 8.075 | 14.297 |
| PAR | 5.706 | 9.189 | 2.771 | 3.992 | 4.326 | 7.516 | 5.016 | 10.18 | 1.554 | 2.592 |
| HSJA | 20.356 | 25.011 | 8.06 | 13.444 | 16.369 | 25.268 | 14.434 | 25.535 | 4.367 | 8.373 |
| PAR+HSJA | 4.752 | 7.781 | 2.388 | 3.719 | 3.644 | 6.688 | 4.517 | 9.093 | 1.522 | 2.51 |
| BBA | 5.849 | 9.069 | 1.643 | 3.125 | 3.692 | 6.422 | 5.423 | 10.875 | 1.263 | 2.315 |
| PAR+BBA | 3.899 | 6.953 | **0.982** | 2.21 | 2.098 | 4.547 | 3.456 | 7.816 | 0.921 | 1.759 |
| Evo | 8.195 | 12.047 | 4.133 | 6.253 | 5.223 | 9.82 | 7.847 | 15.358 | 3.093 | 3.924 |
| PAR+Evo | 4.091 | 7.122 | 2.055 | 3.284 | 2.427 | 5.236 | 4.041 | 8.576 | 1.487 | 2.223 |
| Boundary | 11.25 | 14.102 | 4.8 | 6.068 | 7.963 | 11.533 | 8.047 | 13.583 | 2.442 | 3.876 |
| PAR+Boundary | 4.762 | 8.073 | 2.145 | 3.34 | 3.535 | 5.888 | 4.604 | 8.795 | 1.296 | 2.307 |
| SurFree | 6.331 | 10.485 | 1.505 | 3.486 | 3.048 | 7.849 | 5.979 | 11.001 | 0.949 | 2.25 |
| PAR+SurFree | 4.078 | 6.989 | 1.224 | 2.589 | 2.183 | 4.603 | 4.015 | 7.959 | 1.008 | 1.912 |
| CAB | 4.214 | 8.034 | 1.966 | 3.978 | 2.364 | 10.554 | 3.646 | 12.058 | 1.121 | 2.084 |
| PAR+CAB | **1.963** | **4.879** | 1.012 | **1.824** | **1.244** | **3.484** | **1.752** | **6.145** | **0.694** | **1.423** |
| Sign-OPT | 30.581 | 36.062 | 19.56 | 22.152 | 29.566 | 38.994 | 20.952 | 38.496 | 6.392 | 12.083 |
| PAR+Sign-OPT | 4.525 | 8.067 | 2.602 | 3.73 | 3.578 | 6.679 | 4.91 | 9.387 | 1.353 | 2.548 |

Table 3: Median and average $\ell_2$ distance of adversarial perturbations on ILSVRC-2012.

| Target | res-101 | | dense | | vgg-19 | | senet | | r26_s32 | | vit_s16 | |
|---|---|---|---|---|---|---|---|---|---|---|---|---|
| Methods | Mid | Avg | Mid | Avg | Mid | Avg | Mid | Avg | Mid | Avg | Mid | Avg |
| Initial | 58.60 | 54.71 | 54.38 | 52.77 | 34.80 | 34.67 | 49.52 | 53.96 | 94.72 | 88.49 | 96.25 | 92.94 |
| PAR | 9.19 | 10.60 | 10.08 | 11.50 | 6.25 | 7.09 | 7.80 | 10.80 | 14.71 | 28.52 | 15.10 | 30.75 |
| HSJA | 35.13 | 36.29 | 30.22 | 32.58 | 17.75 | 20.85 | 29.38 | 34.69 | 63.67 | 63.81 | 65.81 | 67.41 |
| PAR+HSJA | 8.92 | 10.06 | 7.97 | 10.84 | 5.89 | 6.75 | 7.76 | 10.06 | 13.75 | 27.42 | 14.55 | 30.05 |
| BBA | 9.00 | 9.78 | 9.17 | 10.83 | 5.44 | 6.85 | 8.72 | 12.31 | 13.16 | 26.88 | 13.86 | 28.36 |
| PAR+BBA | 4.89 | 7.24 | 5.39 | 7.78 | 3.26 | 4.77 | 5.45 | 7.15 | 9.88 | 23.14 | 8.83 | 25.29 |
| Evo | 12.53 | 13.89 | 11.91 | 13.59 | 8.71 | 11.28 | 9.94 | 12.11 | 23.83 | 34.28 | 24.21 | 37.96 |
| PAR+Evo | 6.83 | 8.34 | 5.71 | 8.46 | 4.88 | 5.87 | 5.51 | 7.94 | 11.80 | 24.79 | 11.50 | 27.54 |
| Boundary | 16.25 | 18.17 | 14.78 | 17.65 | 8.96 | 10.92 | 14.53 | 18.29 | 28.05 | 37.54 | 21.52 | 36.53 |
| PAR+Boundary | 7.29 | 9.19 | 6.98 | 9.60 | 4.82 | 5.90 | 6.71 | 8.60 | 11.62 | 25.68 | 11.09 | 27.81 |
| SurFree | 12.27 | 14.57 | 8.99 | 14.07 | 5.05 | 8.02 | 6.59 | 13.52 | 20.35 | 31.56 | 15.88 | 32.09 |
| PAR+SurFree | 6.08 | 7.88 | 5.54 | 8.22 | 3.31 | 4.88 | 4.91 | 7.26 | 10.11 | 23.90 | 9.79 | 26.32 |
| CAB | 11.20 | 19.52 | 9.29 | 19.02 | 3.95 | 9.07 | 9.54 | 22.14 | 18.84 | 38.26 | 18.69 | 44.85 |
| PAR+CAB | **4.73** | **6.86** | **4.05** | **7.76** | **2.51** | **4.29** | **3.81** | **7.05** | **7.07** | **22.60** | **6.19** | **24.96** |
| Sign-OPT | 50.18 | 47.92 | 32.71 | 46.63 | 28.38 | 30.25 | 44.37 | 47.59 | 83.99 | 81.34 | 83.47 | 83.78 |
| PAR+Sign-OPT | 9.16 | 9.63 | 10.02 | 11.14 | 4.37 | 6.43 | 6.80 | 9.63 | 13.16 | 28.23 | 14.23 | 28.78 |

inception-resnet v2 [37], nasnet [38], densenet-161 [39], vgg19-bn [40], senet-154 [41]. 4 RTX 3090 GPU cards are used for calculation.

We compare 7 decision-based attacks with our PAR under the black-box setting with limited query number: Boundary Attack [15], Biased Boundary Attack (BBA) [20], Evolutionary Attack (Evo) [19], HSJA [30], CAB [31], Sign-OPT [33], and SurFree [32]. Stepsizes of spherical direction and source direction are $\delta_0 = 0.1, \varepsilon_0 = 0.003$ for Boundary, BBA, Evo and CAB. For BBA [20], we use the version that does not incorporate information from a substitute model at each step for fair comparison. Noise magnitude limit $\tau = [0, 255]$. The initial and minimum patch size for ImageNet-21k and ILSVRC-2012 is set to 56 and 7, respectively. For Tiny-Imagenet, we set the initial and minimum patch size as 16 and 2, respectively. All datasets are under BSD 3-Clause License.

For evaluation criterion, we choose the median and average size of adversarial perturbation, as applied in NIPS 2018 Adversarial Vision Challenge [28]:

$$mid = median(\{\|x' - x\|_2 \mid x \in \mathbf{X}\}), \quad avg = \frac{1}{n_{data}} \sum_{i=1}^{n_{data}} (\{\|x'_i - x_i\|_2 \mid x \in \mathbf{X}\}), \quad (10)$$

where $n_{data}$ is the number of images in a dataset, $x$ is an original image in the dataset $\mathbf{X}$. $x'$ is the adversarial example found that is closest to $x$. A smaller $\ell_2$ distance indicates a better adversarial example. It is worth noting that adversarial examples are rounded before being input to the target model for a more realistic black-box attack setting.

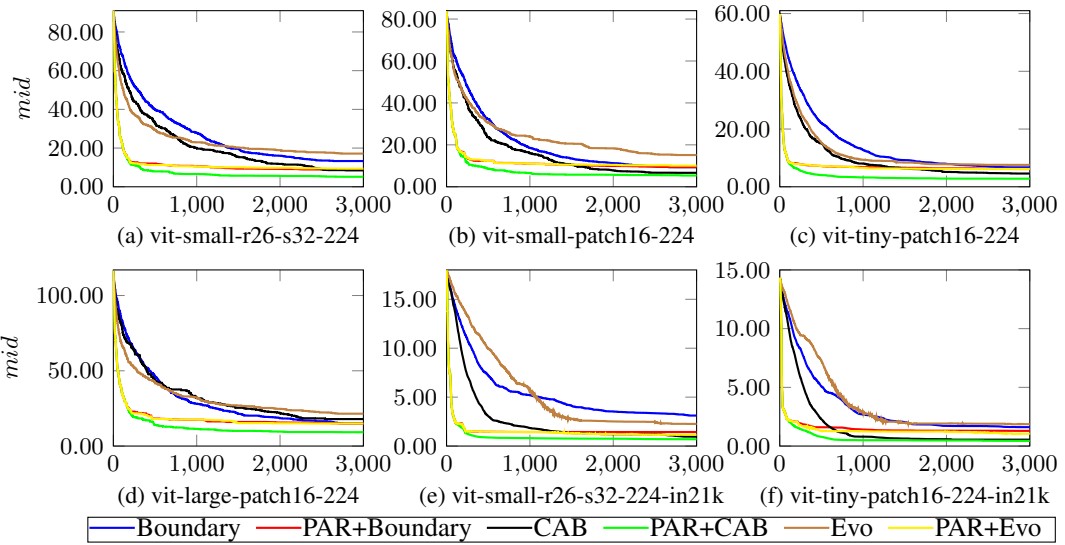

Figure 4: Median $\ell_2$ distance of adversarial noise under different query number $T$.

Table 4: Targeted adversarial perturbations on ILSVRC-2012.

|     | Initial | PAR    | HSJA   | BBA    | Evo    | Boundary | SurFree |
|-----|---------|--------|--------|--------|--------|----------|---------|
| Mid | 152.296 | **39.821** | 92.183 | 67.728 | 69.397 | 52.584   | 57.808  |
| Avg | 154.797 | **40.792** | 93.767 | 70.01  | 69.039 | 51.272   | 55.378  |

## 4.2 Experimental Results

To verify the advantage of PAR over existing decision-based attacks on ViTs and CNNs, we report the median and average adversarial perturbation on ImageNet-21k, ILSVRC-2012, and Tiny-Imagenet in Table 2, Table 3, and Table 1, respectively. The first row of three tables represents target models with different structures. We compare the average (Avg) and median (Mid) noise magnitude generated by PAR and other 6 attacks on different target models. We also use PAR as the noise initialization for other decision-based attacks. The noise compressed by PAR is handed over to other decision-based attacks for further compression. It can be seen that when PAR is used to initialize adversarial noise, the average and median noise magnitude drops significantly compared with only using the original decision-based attack. This verifies the strong noise compression ability of PAR. We also combine PAR with other decision-based attacks, and compare the query efficiency under total query numbers of 3000 in Fig. 4. The target models are notated under each subfigure. There is a noticeable drop in noise magnitude when initializing the noise with PAR.

Table 5 compares the noise compression efficiency and the average query number of PAR under different initial patch sizes and minimum patch sizes. The compressed noise is handed over to Boundary Attack for further compress until 1000 queries. It can be seen that when the initial patch size is small, the average query number will be large, resulting in low query efficiency and less query number for subsequent decision-based attacks. A more reasonable strategy is to use a large initial patch size and stop at a small minimum patch size.

We also extend PAR to targeted attack. We randomly choose an image of target class as starting point and keep the adversarial examples in target class. The target model is vit-small-r26-s32. Targeted results are shown in Table 4. The targeted noise of PAR is still significantly smaller than others.

Fig. 5 compares adversarial noises generated against vit-small-r26-s32-224 by seven different attacks on ILSVRC-2012. The first row shows the original images. The second to eighth images of each row are the noises generated by each attack. PAR stops when the exit condition is met. All other attacks perform 1000 queries on the target model for each adversarial example. The noises of PAR mainly concentrate on a few patches instead of spreading over the entire image. The noise magnitude of other decision-based attacks decrease significantly when PAR is used for noise initialization. PAR

Table 5: Noise compression comparison on different initial patch sizes and minimum patch sizes.

| | Initial Patch Size | 112 | 112 | 112 | 112 | 56 | 56 | 56 | 28 | 28 | 14 |
|---|---|---|---|---|---|---|---|---|---|---|---|
| | Minimum Patch Size | 7 | 14 | 28 | 56 | 7 | 14 | 28 | 7 | 14 | 7 |
| vgg-19 | Mid Noise | **4.31** | 5.07 | 5.55 | 6.21 | 4.34 | 4.88 | 5.54 | 4.60 | 5.09 | 4.79 |
| | Avg Noise | **5.83** | 7.11 | 8.17 | 8.84 | 5.92 | 7.20 | 8.33 | 6.11 | 7.43 | 6.47 |
| | Avg Query Number | 195.69 | 97.30 | 44.54 | 16.80 | 202.98 | 100.88 | 45.79 | 238.24 | 130.58 | 415.06 |
| vit_s16 | Mid Noise | 8.76 | 9.32 | 9.54 | 10.35 | **8.62** | 9.17 | 9.67 | 9.01 | 9.88 | 9.17 |
| | Avg Noise | 17.24 | 19.08 | 19.96 | 20.52 | **17.08** | 18.84 | 19.69 | 17.43 | 19.16 | 17.90 |
| | Avg Query Number | 249.34 | 122.81 | 49.53 | 17.04 | 247.01 | 120.93 | 49.90 | 289.67 | 153.07 | 448.60 |

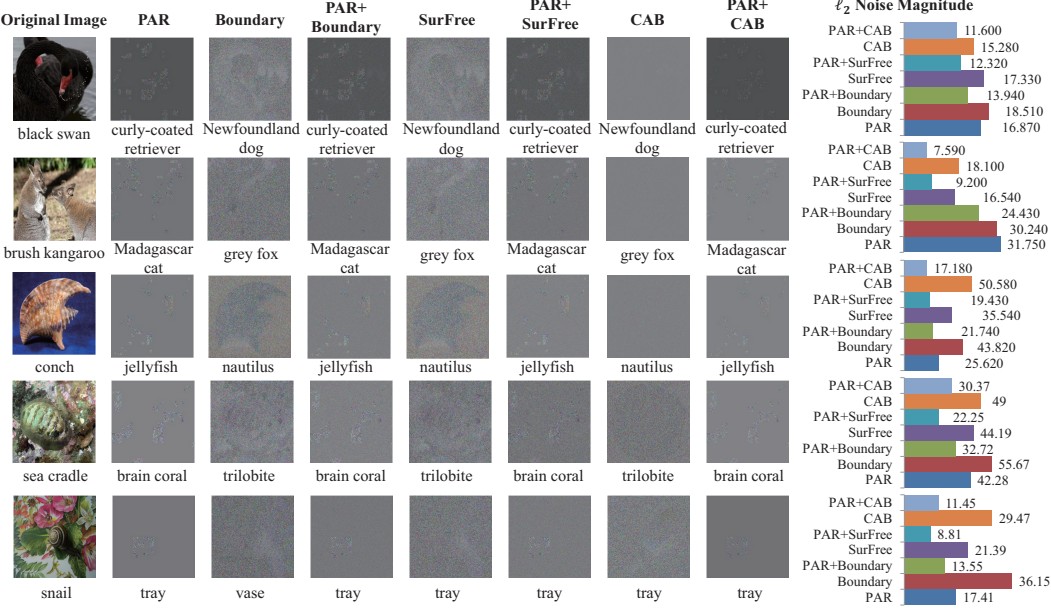

Figure 5: Comparison of adversarial noises generated by PAR, Boundary, PAR+Boundary, SurFree, PAR+SurFree, CAB, and PAR+CAB on ImageNet dataset. The labels and misclassification categories are noted under original images and adversarial noises. The rightmost column compares the noise magnitude of attacks in $\ell_2$ norm. 1000 queries have been performed for each attack except for PAR.

proposed in this paper is only used for the study of adversarial machine learning and the robustness of ViTs, and does not target any real system. There is no potential negative impact. More experimental results on different target ViT and CNN structures are provided in the Supplementary Material A.3.

## 5 Conclusion

In this paper, we explore decision-based adversarial attacks against Vision Transformers. In view of the huge difference in the noise sensitivity between patches of ViTs, we propose Patch-wise Adversarial Removal to achieve query-efficient noise compression. PAR maintains noise magnitude and noise sensitivity masks to probe and compress adversarial noises in a patch-wise manner, and improve query efficiency through a coarse-to-fine search process on the patch size. Experiments on three image classification datasets verify the feasibility and generalizability of PAR to improve the query efficiency with limited query number.

## Acknowledgements

This work is supported by the National Natural Science Foundation of China (NSFC) (under Grant 61876130, 61932009) and the Open Foundation of Henan Key Laboratory of Cyberspace Situation Awareness (No HNTS2022028).

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
