# OpenReview forum: "Decision-based Black-box Attack Against Vision Transformers via Patch-wise Adversarial Removal"
_NeurIPS.cc/2022/Conference — NeurIPS 2022 Accept_

### Official Review · Reviewer_SDVz · 2022-07-09

**Rating:** 6
**Confidence:** 3
**Soundness:** 3 good
**Presentation:** 4 excellent
**Contribution:** 3 good

**Summary:**

This paper proposes a decision-based black-box attack against vision transformers. Previous decision-based black-box attacks do not consider the noise sensitivity difference among different regions in the images.  In this paper, the authors find that different patches in the images have different noise sensitivity, thus it proposes a new decision-based black-box attack via patch-wise adversarial removal (PAR). PAR divides images into patches through a coarse- to-fine search process, and gradually removes the noise in those patches with low noise sensitivity. Extensive experiments demonstrate that when PAR combined with other SOTA methods, it can obtain smaller noise magnitude under same query numbers.

**Questions:**

1. For the noise sensitivity regions in the images, is it more related to the salient region or the background? Can we use the gradient magnitude to indicate the noise sensitivity regions as done in salient region detection?
2. As the attack process is from coarse to fine, though in the experiments it use the same query number, is the computational efficiency of the proposed method lower?


**Limitations:**

No potential negative societal impact

**Strengths And Weaknesses:**

Strengths:
1. This paper is well-written and easy to follow.
2. The motivation of this paper makes sense and is novel to me. It is reasonable that different regions of the images would have different noise sensitivity. The proposed method is straightforward and effective.
3. The experiments are extensive, it compares with different state-of-the-art methods on different architectures and datasets.
PAR can combine with other decision-based methods and improve the performance.

Weaknesses:
1. In Table 1, it seems that on inc-v3 the median noise of PAR+SurFree is not smaller than SurFree, however, the authors bolded the results, is it typos?
2. It seems that the improvements of PAR+SurFree are not significant when compared to SurFree in Table 1 on Tiny-ImageNet dataset, can the authors give some explanations?
3. In Table 1, 2, 3，It seems that the performance of PAR itself is not better than the SOTA methods.

---

> ### Author Response · Authors · 2022-08-02
> **Response to Reviewer SDVz**
>
> Many thanks for the detailed comments. We have uploaded a revised version with added experiments and corrected typos.
>
> **Q1** On inc-v3 the median noise of PAR+SurFree is not smaller than SurFree, however, the authors bolded the results.
>
> **A** Thanks for the careful check, a typo does exist here. In Table 1, the median noise magnitude of Surfree against inc-v3 is the smallest at 0.627 and should be bolded. We have made corrections in the revised version.
>
> **Q2** It seems that the improvements of PAR+SurFree are not significant when compared to SurFree in Table 1 onTiny-ImageNet dataset.
>
> **A** In decision-based attack, when the initial noise is relatively small, the noise compression performance is relatively limited. **Firstly**, the initial noise magnitude on Tiny-Imagenet is relatively low compared to the other two large-scale datasets. Because the magnitude of noise on a whole patch is small, further compression will be difficult. **Secondly**, Surfree has the smallest noise magnitude compared with other decision-making attacks on Tiny-Imagenet. Therefore, the performance improvement achieved by initializing Surfree with PAR is not particularly significant on Tiny-Imagenet. But on the other two datasets, the performance improvement of PAR+Surfree compared to using Surfree alone is indeed significant.
>
> **Q3** In Table 1, 2, 3，It seems that the performance of PAR itself is not better than the SOTA methods.
>
> **A** The reason why the performance of PAR alone is lower than that of SOTA methods is that, PAR does not use all the of queries, but exits when patch size reaches the minimum patch size, saving the number of queries for subsequent decision-based attacks.
>
> **Q4** For the noise sensitivity regions in the images, is it more related to the salient region or the background? Canwe use the gradient magnitude to indicate the noise sensitivity regions as done in salient region detection?
>
> **A** From the adversarial examples generated in Fig. 5 of the paper, the noise-sensitive areas found by PAR are basically consistent with the objects in the foreground. But this is not a general phenomenon. To a certain extent, the remaining patches after noise removal by PAR can indeed be regarded as patches with higher noise sensitivity (because the noises on patches with lowest noise sensitivity has been removed). However, high noise sensitivity and salient regions may not coincide. In some cases, adding noise to the background can also change the classification result of the image. Therefore, PAR may not be a good choice for salient region detection.
>
>
> **Q5** Is the computational efficiency of the proposed method lower?
>
> **A** Since in decision-making attacks, most time is spent on querying the target model, the computational efficiency of different decision-making attacks is similar. However, the noise compression efficiency of PAR is very high. In Table 6 of the revised supplementary material, we compare the time consumption and noise compression efficiency of PAR and other decision-making attacks on the Imagenet. The target model is r-ti-16. The total number of queries is 1000 times. Among them, the first 50 times are used for generating Gaussian noise to find initial adversarial examples. When PAR is not applied, the next 950 times are all used for decision-based attacks. When initialized with PAR, 60 queries are used for PAR, and then the remaining 890 queries are used for decision-based attack. The experimental results report the total time consumption, number of queries, query time per query and average compression noise per query. As demonstrated in the following table, since the main time-consuming of the query lies in the forward propagation process of the target model, the used time of a single query for each method is similar. But it can be seen that the noise compression efficiency of each decision attack method is improved after initializing with PAR. During the first 60 queries of PAR, the noise compression efficiency is significantly higher than other decision-based attacks, which demonstrates the effectiveness of PAR.
>
> |Methods | Time Cost (s) | Used step | Time Per Query (s) | Noise Compression Per Query |
> | :----: |:----: |:----: |:----: |:----: |
> |PAR |2.22 | 60 | 0.037  | **0.673** |
> |Evo |28.28  | 950  | 0.030  |0.035 |
> |PAR+Evo |27.22  | 950  | **0.029**  | 0.045 |
> |Boundary | 31.37  | 950  | 0.033  | 0.040 |
> |PAR+Boundary | 34.72  | 950  | 0.037  | 0.044 |
> |CAB |36.09  | 950  | 0.038  | 0.044 |
> |PAR+CAB| 40.15  | 950 |  0.042  | 0.047 |

---

### Official Review · Reviewer_zV4y · 2022-07-10

**Rating:** 6
**Confidence:** 2
**Soundness:** 3 good
**Presentation:** 3 good
**Contribution:** 3 good

**Summary:**

This work explores decision-based adversarial attacks against Vision Transformers. In view of the huge difference in the noise sensitivity between patches of  ViTs, the authors propose Patch-wise Adversarial Removal to achieve query-efficient noise compression. PAR maintains noise magnitude and noise sensitivity masks to probe and compress adversarial noises in a patch-wise manner, and improve query efficiency through a coarse-to-fine search process on the patch size. Extensive experiments on three datasets demonstrate that PAR achieves a much lower noise magnitude with the same number of queries.

**Questions:**

please see the weakness

**Ethics Review Area:**

["Privacy and Security (e.g., consent)"]

**Limitations:**

please see the weakness

**Strengths And Weaknesses:**

[Strengths]
1.	This work maintains noise magnitude by using the noise sensitivity of the target model. They validate the effectiveness of PAR on three datasets and 18 different target models, and the experiments are extensive.
2.	The narrative of the paper is clear and the proposed method is effective, which can reduce the noise magnitude a lot.
3.	The vision transformers models are based on patches. Noise sensitivity mask and noise magnitude mask can measure the importance of noise on different patches, and the idea is novel.

[Weaknesses]
1.	The minimum patch size can be one to be added in Table 4, which can be the comparison in the per-pixel setting.
2.	The experiment about time cost can be added to verify the superiority of the PAR.
3.	This method is to reduce the size of the adversarial noise, but in practical applications, reducing the number of queries is a more important goal that needs to be optimized.

---

> ### Author Response · Authors · 2022-08-02
> **Response to Reviewer zV4y**
>
> Many thanks for the detailed comments. We have uploaded a revised version with the supplemented results.
>
> **Q1** The minimum patch size can be one to be added in Table 4,
>
> **A** Thank you for your suggestion. We have added experimental results with a minimum patch size of 1 on the basis of Table 4 in the paper. New results are placed in Table 5 of the revised supplementary material. A minimum patch size of 1 means that PAR will try to remove noise on a single pixel. It can be seen from the results in the following table that using a too small minimum patch size will also lead to low compression efficiency. Because when minimum patch size=1, a single query can only remove noise on a single pixel at most even if it succeeds. At the same time, the number of queries consumed by PAR will also increase sharply with a too small minimum patch size.
>
> | Target Model |        Initial Patch Size |     112   |  56   |  28   |  14   |   7 |
> | :----: | :----: |:----: |:----: |:----: |:----: |:----: |
> ||  Minimum Patch Size  |     1    |    1   |     1  |      1     |   1|
> | vgg-19 |  Mid Noise       | 4.73    | 4.95    | 5.20    | 5.98   |  13.05 |
> |        |  Avg Noise       | 6.32    | 6.31    | 6.55    | 7.05   |  11.31 |
> |        |  Avg Query Number| 810.22  | 811.86  | 835.30 |  882.28 | 945.43 |
> | vit_s16|  Mid Noise       | 8.89    | 8.97    | 9.38    | 11.88  |  24.93 |
> |        |  Avg Noise       | 17.68   | 17.53   | 17.49   | 18.90  |  26.84 |
> |        |  Avg Query Number| 825.60  | 831.32  | 855.66 |  909.22  |969.57 |
>
> **Q2** The experiment about time cost can be added to verify the superiority of the PAR.
>
> **A** Thank you for your suggestion. In Table 6 of the revised supplementary material, we compare the time consumption and noise compression efficiency of PAR and other decision-making attacks on the Imagenet. The target model is r-ti-16. The total number of queries is 1000 times. Among them, the first 50 times are used for generating Gaussian noise to find initial adversarial examples. When PAR is not applied, the next 950 times are all used for decision-based attacks. When initialized with PAR, 60 queries are used for PAR, and then the remaining 890 queries are used for decision-based attack. The experimental results report the total time consumption, number of queries, query time per query and average compression noise per query. As demonstrated in the following table, since the main time-consuming of the query lies in the forward propagation process of the target model, the used time of a single query for each method is similar. But it can be seen that the noise compression efficiency of each decision attack method is improved after initializing with PAR. During the first 60 queries of PAR, the noise compression efficiency is significantly higher than other decision-based attacks, which demonstrates the effectiveness of PAR.
>
> |Methods | Time Cost (s) | Used step | Time Per Query (s) | Noise Compression Per Query |
> | :----: |:----: |:----: |:----: |:----: |
> |PAR |2.22 | 60 | 0.037  | **0.673** |
> |Evo |28.28  | 950  | 0.030  |0.035 |
> |PAR+Evo |27.22  | 950  | **0.029**  | 0.045 |
> |Boundary | 31.37  | 950  | 0.033  | 0.040 |
> |PAR+Boundary | 34.72  | 950  | 0.037  | 0.044 |
> |CAB |36.09  | 950  | 0.038  | 0.044 |
> |PAR+CAB| 40.15  | 950 |  0.042  | 0.047 |
>
>
> **Q3** In practical applications, reducing the number of queries isa more important goal that needs to be optimized.
>
> **A** We fully agree that reducing the number of queries in practical applications is a very important goal for decision-based attacks. In fact, this is the core of PAR, which is designed to reduce the number of queries required to compress the noise to a small magnitude. **Firstly**, according to Fig. 4 of the paper, it can be seen that when using PAR as initialization, there will be a cliff-like drop in the noise magnitude in the early stage of noise compression, which intuitively shows the efficiency of PAR. In fact, using PAR alone has been able to compress the noise to a very low level with a small number of queries. **Secondly**, the efficiency of PAR is also reflected in Table 6 of the revised supplementary material. The noise compression efficiency of PAR is significantly higher than other decision-based attacks.

---

### Official Review · Reviewer_6GeV · 2022-07-11

**Rating:** 5
**Confidence:** 3
**Soundness:** 3 good
**Presentation:** 2 fair
**Contribution:** 3 good

**Summary:**

This paper addresses black-box adversarial attack in a decision-based manner, especially for ViT based classification models. Authors investigate the noise sensitivity of each region of the image, and propose an iterative patch based manner to gradually remove noises from patches with larger noises. Experiments are carried on multiple classification models with different datasets, show that the proposed method can work together with other attacks to reduce the noise.

**Questions:**

* How is proposition 1 reflected in algorithm 1?
* In each step of the loop in algorithm 1, noises (z) in the whole patch is set to 0 and then check if the classification is correct. Is it the best strategy to eliminate all noises here? Is it possible to have some smoother version?
* When using PAR as initialization, what is the ratio of noises usually removed?

**Limitations:**

Authors adequately have addressed the limitations and potential negative societal impact of their work.

**Strengths And Weaknesses:**

Strengths:
+ The proposed method is simple and straightforward to implement and is general enough to be applied on multiple types of classification models.
+ The proposed method is effective to serve as an initialization for query-based adversarial attack, can prune out useless noises quickly according to Figure 4. The proposed method can generally boost other  decision-based attacks.

Weaknesses:
- The proposition 1 in the main paper seems to be wrong, and does not align with the line 39 in the appendix. This causes some confusion when reading the paper. Another confusion is that I am not sure how proposition 1 is used in Algorithm 1 and Figure 3. The definition of noise sensitivity in definition 1 and proposition 1 does not align with the Noise Sensitivity Mask $M_S$ in Figure 3, which is a binary matrix where 0 means possibly higher sensitivity. More discussion and clarification are required to address this issue.
- It's unclear how the proposed method correlates with ViTs, other than that both of them are somewhat related to patches. The proposed method works on both ViTs and traditional CNNs, with similar effectiveness. It's unclear if the "noise sensitivity" story only applies on the ViTs, or applies all the classification models. This weakens the statement in line 40 that "The characteristics of ViTs leaving decision-based black-box attacks against ViTs an open problem".

---

> ### Author Response · Authors · 2022-08-02
> **Response to Reviewer 6GeV**
>
> Many thanks for the detailed comments. We have uploaded a revised version with more detailed discussions and corrected typos.
>
> **Q1** The proposition 1 in the main paper seems to be wrong, and does not align with the line 39 in the appendix.
>
> **A** Thank you for your careful check. Proposition 1 is correct, but there is an excessive ellipsis in Eqn. 7 of Supplementary Material. The corrected proof following Eqn. 7 is listed here. We also correct the proof in Supplementary Material:
>
> For any $0 < \kappa_1 \leq \kappa_2 \leq 1$:
>
> $$
>  P(\frac{z^\prime_{r^\ast, c^\ast}}{x^\prime - x} > \kappa_1) - P(\frac{z^\prime_{r^\ast, c^\ast}}{x^\prime - x} > \kappa_2) = P( \kappa_2 \geq \frac{z^\prime_{r^\ast, c^\ast}}{x^\prime - x} \geq \kappa_1) \geq 0, \\
> $$
>
> $$
> P(\frac{z^\prime_{r^\ast, c^\ast}}{x^\prime - x} < \kappa_2) - P(\frac{z^\prime_{r^\ast, c^\ast}}{x^\prime - x} < \kappa_1) = P( \kappa_2 \geq \frac{z^\prime_{r^\ast, c^\ast}}{x^\prime - x} \geq \kappa_1) \geq 0,
> $$
>
> The equality holds when $\kappa_1 = \kappa_2$. Since the probability that noise compression ratio on at least one pixel exceeds the noise sensitivity $Sens$ increases monotonically with respect to the noise sensitivity on the whole patch, and $Sens(F, x, x^\prime, r_1, c_1, h, w)$ < $Sens(F, x, x^\prime, r_2, c_2, h, w)$, we have:
>
>
> $$
> P( F(x^\prime + z_2^\prime) \neq F(x) | F(x^\prime + z^\prime)=F(x) ) \\
> $$
>
> $$
>     = P(\exists r_2 \leq r^\ast_2 \leq r_2+h \ and \ c_2 \leq c^\ast_2 \leq c_2+w, \frac{z^\prime_{r^\ast_2, c^\ast_2}}    {x^\prime - x} < Sens(F, x, x^\prime, r_2, c_2, h, w)) \\
> $$
>
> $$
>     > P(\exists r_1 \leq r^\ast_1 \leq r_1+h \ and \ c_1 \leq c^\ast_1 \leq c_1+w, \frac{z^\prime_{r^\ast_1, c^\ast_1}}{x^\prime - x} < Sens(F, x, x^\prime, r_1, c_1, h, w))  \\
> $$
>
> $$
>     = P( F(x^\prime + z_1^\prime) \neq F(x) | F(x^\prime + z^\prime)=F(x) ).
> $$
>
> Therefore, $P( F(x^\prime + z_1^\prime) \neq F(x) | F(x^\prime + z^\prime)=F(x) ) < P( F(x^\prime + z_2^\prime) \neq F(x) | F(x^\prime + z^\prime)=F(x) )$.
>
> This problem does not affect the proof ideas and conclusions.
>
>
> **Q2** It's unclear how the proposed method correlates with ViTs, other than that both of them are somewhat related to patches.
>
> **A** Compressing noises in patches is the biggest characteristic that distinguishes PAR from other decision-based attacks, which is specifically optimized for ViTs. **On the one hand**, the structural properties of ViTs lead to the need of larger random noise to initialize the decision attack (as can be seen from the first column of Table 3 in the paper). **On the other hand**, since ViTs use patch as the unit of image processing, the differences in noise sensitivity between patches is much larger than that of CNNs (as can be seen from the noise sensitivity comparison between CNN and ViT in Fig. 2 in the revised supplementary material). Therefore, using existing decision-based attacks does not suit for ViTs. PAR is specifically designed to compress noise patch by patch instead of compressing noises on the entire image as a whole. In this paper, CNNs are added to the experimental comparison to verify that PAR can also be effective for CNNs. Therefore, PAR can also be used as a model-agnostic decision-based attack.
>
>
> **Q3** How is proposition 1 reflected in algorithm 1?
>
> **A** Proposition 1 provides inspiration for the new decision-based algorithm. Proposition 1 indicates that, the compression success probability is higher in the area with low noise sensitivity. Therefore, considering that the noise sensitivity varies between different patches of ViTs, noise compression should be performed on each patch separately. We supplement the explanation after the proof idea of proposition 1 in the paper.
>
>
> **Q4** Is it the best strategy to eliminate all noises here? Is it possible to have some smoother version?
>
> **A** If the noise in a patch is not completely removed in single query, but removed little by little, the noise compression efficiency will be affected. For example, if the binary search strategy is used to compress the noise for each patch, the efficiency is much lower than that of the PAR. The main reason is that the noise sensitivity of many patches is close to 0, and it is most efficient to directly remove the noise on these patches. However, it does not rule out the possibility of improving the noise compression efficiency by designing a smoother strategy.
>
> **Q5** When using PAR as initialization, what is the ratio of noises usually removed?
>
> **A** When using PAR as initialization, the ratio of removed noise mainly depends on datasets and target model. According to Table 1, 2, and 3 in the paper, the initial noise removal rate is approximately 80%, 83%, and 85% for Tiny-Imagenet, Imagenet, and ImageNet-21k, respectively.

---

### Official Review · Reviewer_wgUm · 2022-07-15

**Rating:** 6
**Confidence:** 4
**Soundness:** 4 excellent
**Presentation:** 3 good
**Contribution:** 3 good

**Summary:**

This paper proposes a new decision-based attack for vision transformers (ViTs) with improved query efficiency. In particular, it explicitly considers the difference in noice sensitivity of different patches and queries the low-sensitivity patch first. The evaluation results show the improved query efficiency and smaller perturbation distances over the baseline methods on top of various ViTs and CNNs.

**Questions:**

My questions are listed in the weakness part. I'm willing to adjust my scores if the concerns are properly addressed.

**Limitations:**

Although this paper targets adversarial attacks for ViTs, it can help the community better understand the behaviors of ViTs and build defense methods, thus it does not suffer from an obvious negative societal impact.

**Strengths And Weaknesses:**

**Strength**
1. This paper is the first to target decision-based black-box attacks for ViTs.

2. The proposed method is based on the observation of diverse noice sensitivity of different patches in ViTs with good soundness, which achieves much improved query efficiency over SOTA methods.

3. This paper is clearly written with a coherent logical flow.


**Weakness**
1. Although the motivation about the more diverse noice sensitivity in ViTs than CNNs makes sense, it lacks experimental verification or visualization. For example, whether the $\kappa$ in Eq.(3) and Fig.2 is more diverse in ViTs than those in CNNs? This could decide whether the proposed method is more customized for ViTs than CNNs, or it's generally applicable as a new attack method.

2. It's not clear whether the order of the proposed patch-wise adversarial removal has a high correlation with the noise sensitivity in ViTs. According to [1], "patch" is a special granularity that makes ViT less robust than CNNs, which could also partially explain the query efficiency of the proposed method.

3. The experimental results lack enough explanation and analysis:
- For baseline methods, can they achieve worse query efficiency or larger perturbation distances on ViTs than those on CNNs under a similar model complexity?
- Does the proposed method achieve a more notable improvements on ViTs than those on CNNs under a similar model complexity?
- How does the final removal strategy correlate with the noise sensitivity?

4. Missing reference:

[1] "Patch-Fool: Are Vision Transformers Always Robust Against Adversarial Perturbations?", Y. Fu et al., ICLR 2022.

[2] "Give Me Your Attention: Dot-Product Attention Considered Harmful for Adversarial Patch Robustness", G. Lovisotto et al., CVPR 2022.

[3] "Reveal of Vision Transformers Robustness against Adversarial Attacks", A. Aldahdooh et al., arXiv 2021.

---

> ### Author Response · Authors · 2022-08-02
> **Response to Reviewer wgUm**
>
> **Q1** The motivation about the more diverse noise sensitivity in ViTs than CNNs lacks experimental verification or visualization.
>
> **A** The main motivation of PAR is that there is a huge difference in noise sensitivity between ViTs and CNNs. We have added validation and visualization of this motivation in A.1 of the revised supplementary material. Fig. 2 in the revised supplementary material compares the differences of $\kappa$ between res-101 and r26-32. It can be seen that only removing the noises on a few patches on the CNN will affect the misclassification, while the patch-wise noise sensitivity on ViT varies greatly. Therefore, existing decision-based attacks that treat all pixels equally regardless of their noise sensitivity are not fit for ViTs. PAR is specifically designed to compress noise patch by patch instead of compressing noises on the entire image as a whole. In this paper, CNNs are added to the experimental comparison to verify that PAR can also be effective for more models.
>
>
>
> **Q2** It's not clear whether the order of the proposed patch-wise adversarial removal has a high correlation with the noise sensitivity in ViTs.
>
> **A** The order of removing noise for different patches in PAR is not strictly according to the difference in noise sensitivity of different regions. However, it is precisely because of the difference in noise sensitivities of different patches of ViT that PAR follows a patch-by-patch principle to compress noises. **First of all**, the noise sensitivities between different patches of the target ViT cannot be known before querying in the black-box scenario. Therefore, it may be an unrealistic practice to guide noise compression by ranking the noise sensitivity of patches under limited query number. This is also the reason why PAR attempts to completely remove the noise of a patch-wise. **Secondly**, it can be confirmed that noises on patches that are successfully compressed are patches with relatively low noise sensitivity. **In addition**, the Patch-Fool method designed in [1] attacks against the self-attention mechanism of ViTs. Its effectiveness is based on the prior that the target model is a ViT in a white-box setting and can be back-propagated. PAR focuses on the characteristic that ViTs processes image in patches. This is different from the Patch-Fool attack that focuses on the self-attention mechanism of ViTs. But it is possible that the effectiveness of these two attacks are essentially the same.
>
>
> **Q3** The experimental results lack enough explanation and analysis.
>
> **A** Thanks for your constructive comments. We supplement the discussion of these three parts:
>
> **Q3(1)** For baseline methods, can they achieve worse query efficiency or larger perturbation distances on ViTs than those on CNNs under a similar model complexity?
>
> **A** We take the amount of model parameters as the complexity of the model. According to Table 3 in the paper, resnet-101 is one of the most robust CNNs trained on Imagenet with a parameter size of 45M [2]. The query efficiency and perturbation distances of r26-s32 is much worse than resnet-101. In fact, all 8 ViTs trained on Imagenet except r-ti-16 are more robust than all CNNs . In particular, the robustness of vit-s16 is also very high with rather small parameters, so there is no necessary relationship between model robustness and model complexity in the black-box decision-based attacks.
>
> **Q3(2)** Does the proposed method achieve a more notable improvements on ViTs than those on CNNs under a similar model complexity?
>
> **A** Take the CNN model resnet-101 and the ViT model r26-s32 with similar parameters on Imagenet dataset as example. Without considering the combination of other decision-based attacks, the absolute noise compression value of PAR on the ViT is significantly higher than that of CNN (median distance decrease of 80.01 vs. 49.41). The improvement is also relatively significant when combined with other decision-making attack methods. For example, when used to initialize the CAB, PAR decreases the noise by 6.47 on CNN and 10.77 on ViT.
>
> **Q3(3)** How does the final removal strategy correlate with the noise sensitivity?
>
> **A** **Firstly**, any patch of noise that is not removed by PAR is the noise that will affect the misclassification of the adversarial example after removal. As can be seen from Fig. 5 of the paper, compared to other decision-based attacks, the proportion of retained noise is much smaller compared to the removed noise. **Secondly,** the area of retained noise has a high correlation with the foreground object in the image, which is intuitively the area with high noise sensitivity.
>
>
> **Q4** Missing reference.
>
> **A** Thanks for your kind remind. We include these three references into the related work of the revised paper.
>
> [1] Patch-fool: Are vision transformers always robust against adversarial perturbations? in ICLR, 2022.
> [2] Understanding robustness of transformers for image classification, in ICCV, 2021.

---

> > ### Comment · Reviewer_wgUm · 2022-08-09
> > **Reviewer response**
> >
> > Thanks for providing the detailed explanations. It is good to see "the absolute noise compression value of PAR on the ViT is significantly higher than that of CNN" as well as the newly provided visualizations. Most of my concerns are addressed and I will increase my rating to 6.

---

> > > ### Author Response · Authors · 2022-08-09
> > > **Thank you for re-evaluating our work!**
> > >
> > > We are very grateful for your efforts in re-considering our work and our rebuttal. We would keep improving our manuscript to involve your insights.

---

### Meta-Review · Area_Chair_ATTd · 2022-08-21

**Recommendation:** Accept
**Confidence:** Certain

**Metareview:**

The paper proposed a new decision-based black box attack approach for ViTs. The reviewers appreciate the novelty, extensive experiments and clear writing and unanimously vote for acceptance. Authors' responses helped clarify reviewer concerns and new results, as well as analysis, were presented in the rebuttal. ACs suggest accepting the paper and would request authors to include suggested changes in the final version.

**Award:**

Yes

---

### Decision · Program_Chairs · 2022-09-14

Accept